# Oxytetracycline versus Doxycycline Collagen Sponges Designed as Potential Carrier Supports in Biomedical Applications

**DOI:** 10.3390/pharmaceutics11080363

**Published:** 2019-07-24

**Authors:** Graţiela Teodora Tihan, Ileana Rău, Roxana Gabriela Zgârian, Camelia Ungureanu, Răzvan Constantin Barbaresso, Mădălina Georgiana Albu Kaya, Cristina Dinu-Pîrvu, Mihaela Violeta Ghica

**Affiliations:** 1Faculty of Applied Chemistry and Materials Science, University Politehnica of Bucharest, Polizu Street No. 1, 011061 Bucharest, Romania; 2Department of Collagen, Division Leather and Footwear Research Institute, National Research and Development Institute for Textile and Leather, 031215 Bucharest, Romania; 3Department of Physical and Colloidal Chemistry, Faculty of Pharmacy, University of Medicine and Pharmacy “Carol Davila”, 20956 Bucharest, Romania

**Keywords:** collagen, oxytetracycline, doxycycline, drug release, antimicrobial susceptibility, MTT test

## Abstract

Many research studies are directed toward developing safe and efficient collagen-based biomaterials as carriers for drug delivery systems. This article presents a comparative study of the properties of new collagen sponges prepared and characterized by different methods intended for biomedical applications. The structural integrity is one of the main properties for a biomaterial in order for it to be easily removed from the treated area. Thus, the effect of combining a natural polymer such as collagen with an antimicrobial drug such as oxytetracycline or doxycycline and glutaraldehyde as the chemical cross-linking agent influences the cross-linking degree of the material, which is in direct relation to its resistance to collagenase digestion, the drug kinetic release profile, and in vitro biocompatibility. The enzymatic degradation results identified oxytetracycline as the best inhibitor of collagenase when the collagen sponge was cross-linked with 0.5% glutaraldehyde. The drug release kinetics revealed an extended release of the antibiotic for oxytetracycline-loaded collagen sponges compared with doxycycline-loaded collagen sponges. Considering the behavior of differently prepared sponges, the collagen sponge with oxytetracycline and 0.5% glutaraldehyde could represent a viable polymeric support for the prevention/treatment of infections at the application site, favoring tissue regeneration.

## 1. Introduction

Antimicrobial resistance has become a major problem in recent years, and many studies in medicine are focused on this topic. Antibiotics are used both in human and veterinary therapies for infections reduction, and therefore, the role of resistance monitoring is very important [1,2]. Oxytetracycline hydrochloride (OTC) was discovered in the 1940, and was the first member of the tetracycline group [3]. The other tetracyclines were identified later, such as doxycycline hydrochloride (DXC) [4], which came into human pharmacological use in 1967. Due to their broad spectrum of action, OTC and DXC (Figure 1) are antimicrobial substances with bacteriostatic activity that are indicated for the treatment of some infections caused by Gram-negative and Gram-positive microorganisms [5]. OTC inhibits collagenase, preventing the collagen destruction that occurs at the marginal periodontium, and which also inhibits alveolar bone resorption while having an antibacterial effect. Generally, tetracyclines are effective against most spirochetes and anaerobic bacteria; this action is relevant in terms of periodontitis [6,7,8,9].

Considerable attention has focused on the development of drug delivery systems, and this is expected to continue in the future. Drug delivery is a method that is used in the administration of pharmaceutical compounds to achieve a therapeutic effect [10,11,12]. One of the most common approaches to achieve controlled release is to load a drug in a hydrophobic or hydrophilic matrix in order to accelerate or decrease the drug release, depending on the final applications. To obtain a reliable and efficient matrix-based delivery system, many aspects such as biocompatibility, polymers science, drug–matrix interactions, and drug physicochemical and therapeutic characteristics are required to merge together. An optimal drug delivery system can really have the desired benefit only if the drug is delivered at the targeted site and maintains performance for the proper duration.

A wide variety of materials classified in natural and synthesized biodegradable polymers have been used to develop delivery systems [13]. The first category includes collagen (CG) [14], gelatin, chitosan, and alginate, which are biocompatible and biodegradable. Collagen is a highly versatile material, and due to its well-established safety profile, it represents a favorable matrix to obtain carrier systems for the delivery of drugs, proteins, and genes [15,16,17,18,19], or generally to obtain composite scaffolds for bone tissue engineering [20,21]. Depending on the final therapeutic target, the collagen is treated by several methods such as physical [22], chemical [23,24] or enzymatic cross-linking [25,26,27]. Numerous methods related to the cross-linking of collagen-based biomaterials are described in the literature [28], with aldehydes such as glutaraldehyde (GA) and formaldehyde being widely used over the last decade. GA is a very reactive, available, and inexpensive chemical agent [29].

In our previous works, we successfully used glutaraldehyde as cross-linking agent for the collagen biomaterials processed in various forms (hydrogels, sponges) [11,21,24,30,31,32,33]. The physical–chemical, biopharmaceutical, and biological properties of the prepared drug-loaded collagen supports were improved in the presence of GA in terms of resistance to enzymatic degradation, goniometric characteristics, and morphological characteristics, ensuring at the same time controlled drug release profiles over several hours. Moreover, the designed carriers have maintained the integrity and the preservation of the collagen triple helix, and also have ensured the support viability and proliferation of different tested culture cells. Thus, even glutaraldehyde is contested as being a toxic cross-linking agent; we proved in our previous studies about collagen and doxycycline that GA is accepted by endothelial cells when it is used in small amounts [34]. 

The effect of the cross-linking agent and drug on the biostability of the treated biopolymeric material is very important [35]. Due to their intrinsic biodegradability, biomaterials are digested in body fluids. For the in vivo stability evaluation of the collagen-based biomaterials, the in vitro enzymatic degradation is studied. The collagen degradation process requires the presence of water and enzymes, and digestion of the bonds. By exposure to water, collagen swells, but complete collagen degradation occurs only when enzymes are involved. Very few enzymes are able to attack and degrade the triple helical structure of collagen. The enzymes that break down collagen into small peptides are called collagenases [36]. These enzymes have the ability to cleave collagen at physiological pH and temperature. Collagenases are produced by bacteria such as *Clostridium hystolyticum*, *Pseudomanas aeruginosa*, *Bacteroides melanogenicus*, *Mycobacterium tuberculosis*, and molds such as *Streptomyces madurae*, *Trichophyton schoenleinii,* and *Aspergillus oryzae*. Collagenase may simultaneously split each polypeptide chain across all three chains or attack at a single strand.

The collagen sponge morphological structure—pore size and porosity are two key important parameters, influencing the hydrophilicity, drug diffusion through the network, degradation properties, and interactions with cells. Collagen-doxycycline and collagen-minocicline cross-linked with glutaraldehyde sponges with different pore sizes were obtained by different temperatures of freezing during lyophilization, and the results showed that the sponges’ composition as well as their preparation process influenced the swelling behavior, the released drug percentage, and the porous architecture [37,38]. 

In this study, a therapeutic agent such as oxytetracycline or doxycycline is mixed in a polymeric matrix and treated with GA. As final products after a lyophilization process, sponges with different compositions were obtained and characterized by different experimental tools and techniques to establish the physicochemical characteristics. Additionally, the influence of matrix composition changes on drug release kinetics and biostability was evaluated. From the antimicrobial susceptibility point of view, antibiotic resistance of collagen matrix to *Escherichia coli*, *Staphylococcus aureus*, and *Enterococcus faecalis* was studied. Moreover, the effect of combining collagen with OTC or DXC on the enzymatic degradation and in vitro biocompatibility with dermal fibroblasts-type National Collection of Type Cultures (NCTC) was tested.

## 2. Materials and Methods 

### 2.1. Materials and Preparation of Collagen Sponges 

Collagen type I extracted from bovine skin was supplied by National Research & Development Institute for Textiles and Leather, Bucharest, Romania. The antimicrobial drugs oxytetracycline hydrochloride and doxycycline hydrochloride were purchased from Sigma Chemical Co. (Saint-Louis, MO, USA), and glutaraldehyde was used as the cross-linking agent and purchased from Merck (Darmstadt, Germany). For pH adjustment, 1 M of NaOH solution of analytical grade was used. 

Collagen supports preparation, in the form of sponges, was based on a freeze-drying process [30] of homogenous mixtures containing CG, OTC, or DXC and GA. CG 1% with pH adjusted to 7.4 was mixed with a water solution of OTC or DXC 1 g/L for 30 min, and after another pH adjustment to 7.4, different concentrations of GA reported to dry CG and OTC or DXC substances were added and stirred for 10 min. The resultant mixtures were then freeze-dried. This process consists of three steps as follows: “Freezing” at −55 °C for 10 min, “Main drying” at 55 °C for 15 h, where ~90% of water is removed, and “Final drying” at 40 °C for 10 min. All the collagen sponges were packed in polyethylene bags and sterilized for 5 min on each side at 254 nm using a Vilber-Lourmat equipment.

The composition of the collagen sponges is shown in Table 1.

### 2.2. Methods

Infrared spectroscopy (IR) measurements were performed using a Perkin Elmer Spectrum 100 FTIR spectrophotometer (Beaconsfield, UK) with a diamond ATR (attenuated total reflectance) device attached. The ATR-FTIR spectra were obtained over the domain of 4000–600 cm^−1^ at a spectral resolution of 4 cm^−1^. For each point, 16 scans were collected at 25 °C.

UV/Vis/NIR spectroscopy was performed on a Jasco UV/Vis/NIR spectrophotometer (Easton, MD, USA), model V 670. The spectra were recorded over the wavelength range of 200 to 2000 nm, with a step of 0.5 nm.

The swelling behavior of the collagen supports was evaluated by testing their ability to absorb water. The collagen sponges were cut into pieces of 0.5 × 0.5 cm and weighed before immersion in distilled water (*W_i_*) for different periods of time up to 5 h at 25 °C. After that, the swollen collagen sponges were removed and weighed without dripping (*W_f_*). The swelling capacity was calculated using Equation (1). Each test was averaged from three parallel measurements.
(1)$$\%\;S\;=\;\frac{w_{f}-w_{i}}{w_{i}}\;\times \;100$$

The enzymatic degradation was carried out by exposing in vitro the collagen biomaterials to a collagen specific enzyme. Bacterial collagenase of *Clostridium histolyticum* from Sigma-Aldrich (Saint-Louis, MO, USA) and phosphate buffer solution (PBS) (pH 7.4) were used. To test the in vitro degradation, all the samples were cut into pieces of 0.5 × 0.5 cm, weighed before transfer in PBS (*W_i_*), and incubated at 37 °C for 24 h. Then, collagenase (10 μg/mL) was embedded, and each sample was incubated again at 37 °C. At various time intervals (1 h, 2 h, 4 h, 8 h, 24 h, 48 h, 72 h, 96 h, and 168 h), the sponges were removed from the solution, squeezed, and weighed (*W_t_*). The calculation of collagen degradation was performed as follows (Equation (2)):(2)$$\%\;collagen\;degradation=\;\frac{w_{i}-w_{t}}{w_{i}}\;\times \;100$$

Three samples were evaluated for each composition and the results of collagen degradation are expressed as mean values ± standard deviation.

Antimicrobial assay was tested against three main human pathogenic bacteria: Gram-negative bacterium, *Escherichia coli* ATCC 8738, respectively, Gram-positive bacteria, *Enterococcus faecalis* ATCC 29212 (facultative anaerobe), and *Staphylococcus aureus* ATTC 25923. The stock culture was maintained at 4 °C.

These strains were cultivated onto Luria–Bertani liquid medium (abbreviated “LB”) having the following composition: peptone (Merck, Darmstadt, Germany), 10 g/L; yeast extract (Biolife, Bucharest, Romania) 5 g/L, NaCl (Sigma-Aldrich, Saint-Louis, MO, USA) 5 g/L, and for bacto-agar (for plates, LBA, Sigma-Aldrich, Saint-Louis, MO, USA) 20 g/L agar (Fluka, Bucharest, Romania). LB was sterilized by autoclaving for 20 min at 15 psi (1.05 kg/cm^2^) on liquid cycle. 

The inhibition percentage of bacteria growth (as a measure of antibacterial activity) was calculated using the formula proposed by Jaiswal et al. in 2010 [39,40].

All the collagen samples evaluated for antibacterial activity were firstly sterilized with Vilber Lourmat equipment, Marne-la-Vallée, France. 

Minimum bactericidal concentration (MBC) of oxytetracycline and doxycycline against the previously mentioned bacterial strains was determined using the broth dilution technique [41].

The MBC (defined as 100% killing) represents the lowest concentration of antibiotic that completely inhibited the bacteria growth. Serially diluted logarithmic concentrations of the antibiotic used, ranging from 400 μg/mL to 0.195 μg/mL, were inoculated with standardized overnight cultures of the bacteria and incubated at 37 °C for 18 h [42]. All the data were expressed as the mean ± standard deviation (SD) by measuring three independent replicates. Standard deviation was calculated as the square root of variance using the STDEV function in *Excel* 2010.

In vitro drug delivery from collagen sponges and kinetic analysis of the release data. The kinetic experiments were performed using a paddle dissolution equipment (Esadissolver, Milan, Italy) fitted with a sandwich device. The CG–OTC and CG–DXC sponges were placed on the sandwich device surface and immersed in phosphate buffer solution (pH 7.4). The release medium was maintained at 37 °C and continuously stirred at 50 rpm during the in vitro drug delivery evaluation. Samples of 5 mL were withdrawn from the release medium at specific time intervals over 10 h for CG–OTC sponges, and 8 h for CG–DXC sponges, respectively, and replaced by an equivalent volume of fresh, preheated phosphate buffer solution. The concentration of released antibiotic from the collagen sponges was monitored by UV-Vis spectrophotometry (Perkin Elmer UV-Vis Spectrophotometer, Überlingen, Germany) and determined from the standard curve obtained in the phosphate buffer solution pH 7.4, at wavelength of 363 nm for OTC ($A_{1\%}^{1\mathrm{cm}}=262$, *R*^2^ = 0.9993) and 348 nm for DXC ($A_{1\%}^{1\mathrm{cm}}=204$, *R*^2^ = 0.9979).

To analyze the antibiotics release kinetics from collagen sponges, the in vitro experimental data were fitted according to power law kinetic model (Equation (3)): (3)$$\frac{m_{t}}{m_{\infty }}=k\times t^{n},$$
where *m_t_*/*m*_∞_ represents the fraction of drug released at time *t*; *k*—the kinetic constant related to the characteristics of the drug delivery system expressed in 1/min^n^; and *n*—the release exponent indicating the drug kinetic release mechanism [30]. Two particular cases of this equation could appear: (i) *n* = 0.5 (Higuchi model) corresponding to a Fickian diffusion mechanism, the release being due to the drug diffusion, and (ii) *n* = 1 indicating a zero-order kinetics, and the release involving the polymer relaxation. Values of *n* are associated with an anomalous drug transport mechanism, with different physicochemical processes being involved in the delivery process.

For measurements of cytotoxic activity, fragments of sponges were maintained for 24 h and 48 h respectively, in direct contact with the cell culture, followed by thiazolyl blue tetrazolium bromide (MTT) cytotoxicity test for both time intervals and by an optical microscopy examination of cell morphology after 48 h, as highlighted by hematoxylin–eosin staining. For testing the in vitro cytotoxicity of the sponges, dermal fibroblasts of mice cell line NCTC (clone 929) ECACC, Sigma-Aldrich were grown in a monolayer in minimum essential medium (MEM) with 10% fetal bovine serum (FBS) and 1% mixture of antibiotics (penicillin, streptomycin, and neomycin—PSN) were used. The cell culture, which was subcultured every three days, was kept under humid atmosphere with 5% CO_2_ at 37 °C. The experiment was performed in triplicate. Sponges with an area of 0.25 cm^2^ (per 0.5 mL solution) were sterilized by UV radiation for 8 h.

Cells seeding was performed at a cell density of 4 × 10^4^ cells/mL determined by the device type hemocytometer with a Carl Zeiss inverted microscope, Thornwood, NY, USA, and a Revco incubator operating at 37 °C, 5% CO_2_ in humid atmosphere. After an incubation time of 24 h, the culture medium was replaced by fresh culture medium (MEM with PSN and 10% FBS) except for the positive control (M+) where 0.003% of hydrogen peroxide solution was added in the culture medium. Fragments of sponges were placed in the culture medium over the cells in corresponding wells of the samples (one fragment/well). After incubation, the solutions and samples were removed from the wells and replaced with 0.25 mg/mL of MTT solution and incubated for 3 h. Then, the solutions from the wells were replaced by isopropanol to dissolve the formazan crystals produced in the cells. The color uniformity of the solution and dissolution of all the formazan crystals were achieved by shaking on an orbital shaker for 15 min. The absorbance was determined with the spectrophotometer microplate reader Berthold Mithras LB 940 multimodal for a lamp power of 40,000 Ǻ and a wavelength of 570 nm. 

Statistical analysis. All the experiments performed on the collagen sponges studied were carried out in triplicate, and the data obtained were analyzed for statistical significance using analysis of variance (one-way ANOVA) and Tukey’s test to determine the significant differences between means. Significant differences were considered for *p* values < 0.05. The results were reported as samples mean ± standard deviation (SD) of independent replicates.

For cell morphology examination with an optical microscope was used. All of the chemicals needed (eosin, hematoxylin, saturated aqueous solution of picric acid, glacial acetic acid, glycerol, alum potassium, potassium iodate, PBS, deionized water, tap water, formaldehyde, calcium chloride) were obtained from Sigma-Aldrich. The steps of cell seeding and the addition of the samples after 24 h were carried out similarly to those used in the MTT method. Two pieces of sponge (5 × 5 mm^2^) and 1000-μL working solutions were added to each well. In the positive control well (M+), the hydrogen peroxide solution of 0.003% prepared in the culture medium was added. After 48 h, the samples from the culture plates and the culture medium maintained in an incubator in humidity conditions, at constant temperature (37 °C) were removed and washed twice with PBS. The PBS was removed, and the cells were fixed for 10 min with 1 mL of Bouin/well. Then, the Bouin solution was removed and the cells were washed three times with deionized water. After the deionized water removal, 1 mL/well of hematoxylin solution was added, left for 5–7 min, and removed, followed by two consecutive washes with running water and one wash with deionized water. The water was removed and eosin solution (1 mL/well) was added, left for 5 min, and removed. After five consecutive washes with deionized water, the plates were dried on a filter paper. Finally, the cell morphology was visualized with an optical microscope.

## 3. Results

### 3.1. Infrared Spectroscopy (IR) Measurements

To detect the specific vibrations of chemical functional groups for all the raw materials and for the collagen sponges, infrared spectroscopy was used. The IR spectra of OTC and DXC powder are presented in Figure 2, and the main IR frequencies of the functional groups are shown in Table 2. 

FT-IR measurements of sponges with collagen and different concentrations of GA [30] with and without antimicrobial drug were also carried out to illustrate the chemical composition (Figure 3). All the collagen-based biomaterials prepared showed the typical absorption bands corresponding to collagen and the antimicrobial drug, as well as to the hydrogen bonds formed between the components [43]. More information [44,45] about the collagen sponges obtained can be extracted from IR spectra as follows: (a) hydrolysis degree calculated as the ratio of the band intensities A_OH_/A_I_ (A_~3300_/A_~1630 cm^−1^_); (b) cross-linking degree obtained by the ratio of the band intensities A_I_/A_OH_ (A_~1630_/A_~3300 cm^−1^_), high values of this ratio indicating a high cross-linking degree; (c) denaturation process evidenced by Equation (4), where $\nu_{I}$ and *ν_II_* represents the wavenumbers of Amide I and Amide II, respectively; for Δ*ν* < 100, no distortion takes place.
(4)$$\Delta \nu =\nu_{I}-\nu_{II}$$

The IR spectra interpretation [43] is highlighted in Table 3 [46].

### 3.2. UV/Vis/NIR Spectroscopy Measurements

UV/Vis/NIR spectroscopy was used to depict the major components of the collagen-based biomaterials with antimicrobial drug and different concentrations of GA. All the sponges were scanned in reflection mode, and the UV/Vis/NIR spectra are presented in Figure 4. 

Table 4 summarizes the results of the reflection UV/Vis/NIR measurements evidencing the main absorption bands specific to the raw materials, and also to the new bonds formed between components as follows: the band in the 223–313 nm domain, which is attributable to n → π * of –CO–NH– from the amide structure of CG; the band corresponding to CH_2_ in the 1181–1189-nm domain; a peak specific to ν_OHassociated_ around 1500 nm; and an intense absorption peak assigned to δ_O-H_ registered between 1937–1953 nm.

### 3.3. Swelling Capacity of the Collagen Sponges

The swelling capacity of a matrix, depending on the number of hydrophilic groups [47], is a required characteristic for biomedical applications such as dressings. Each dried collagen support was swollen in distilled water at room temperature to establish the water absorption capacity, and the standard deviations were labeled with error bars calculated from three samples. The swelling capacity of the sponges is presented in Figure 5, where the water absorption maximum for each sample is noticed. The maximum water absorption for collagen sponges with GA and OTC was recorded at 5 h, while for samples with DXC, the maximum was recorded at 4 h.

### 3.4. Enzymatic Degradation 

The biomaterial structural integrity is an important characteristic when its easy removal from the treated area is required. In vitro tests for the biostability of collagen-based biomaterials illustrated an improvement of their resistance to collagenase digestion due to the presence of antibiotics, and also due to the cross-linking process used in the sponges’ preparation (Figure 6 and Figure 7). 

### 3.5. Antimicrobial Assay

The antimicrobial susceptibility assay of oxytetracycline on the tested bacteria, MBCs of 100 μg/mL, 12.50 μg/mL, and 200 μg/mL against *Escherichia coli*, *Staphylococcus aureus*, and *Enterococcus faecalis*, respectively is presented in Table 5.

In three days, the antimicrobial susceptibility was determined, and MBC values, as the average of three experiments, were reported for each isolate.

In Table 6, the results obtained for the antimicrobial susceptibility test of DXC on the selected bacteria showed MBCs of 200 μg/mL, 100 μg/mL, and 200 μg/mL against *Staphylococcus aureus*, *Escherichia coli*, and *Enterococcus faecalis*, respectively.

The inhibition percent of bacteria growth for the system with OTC is presented in Figure 8. 

The inhibition percent of bacteria growth for the doxycycline collagen sponges is shown in Figure 9.

### 3.6. In Vitro Drug Release Study

The antibiotics delivery profiles represented as cumulative drug-released percentage versus time are presented in Figure 10 for CG–OTC and in Figure 11 for CG–DXC. 

To evaluate the kinetics of drug release as a function of cross-linking agent concentration, the experimental data are presented in Table 7 and Table 8.

### 3.7. Cytotoxic Activity

For in vitro cytotoxicity analysis, both the quantitative method by MTT test and the qualitative method for cell analysis by hematoxylin–eosin staining were used in parallel. Considering the results from the antimicrobial activity, enzymatic degradation, and swelling behavior, for the cytotoxic test, the collagen sponge containing OTC and 0.5% GA was chosen and compared with the collagen sponges non-treated or treated with GA (0% and 0.5% GA).

The cell viability was determined with Equation (5), and the results are shown in Figure 2.
(5)$$\%\;cell\;viability=\frac{Absorbance\;value\;sample}{Absorbance\;value\;control}\times 100$$

## 4. Discussion

According to the IR data (Figure 3), the absorption band attributed to ν_NHas_ and ν_OHas_ at 3294 cm^−1^ for the sponge based just on collagen [12] is recorded at higher wavenumbers in the collagen sponges with the drug and 0.25% GA (3297 cm^−1^ for CG:OTC:GA 0.25%, 3300 cm^−1^ for CG:DXC:GA 0.25%). This shift is attributed to the association degree increasing by hydrogen bonding formed between molecules of collagen and the drug. The amide I band, recorded at 1630 cm^−1^ in the collagen sponge [12] and assigned to the stretching vibration of the C=O (ν_c=o_), is found in the sponges containing CG, the drug, and GA (CG:OTC:GA 0.5%, and CG:DXC:GA 1%) at the same wavenumbers or slightly moved to higher wavenumbers, indicating the association by hydrogen bonding, which increased in the presence of GA (1629 cm^−1^ for CG:OTC:GA 0.75%, 1630 cm^−1^ for CG:OTC:GA 1%, 1631 cm^−1^ for CG:OTC:GA 0.25%, CG:DXC:GA 0.5% and CG:DXC:GA 0.75%, 1632 cm^−1^ for CG:DXC:GA 0.25%, 1634 cm^−1^ for CG:OTC:GA 0.5% and CG:DXC:GA 1%). The amide II band, recorded at 1549 cm^−1^ in collagen sponge [12], is observed at the same wavenumbers in the sponges based on CG and OTC with or without GA. In the biomaterial with CG and DXC embedded, a wavenumber decrease was noticed (1540 cm^−1^ for CG:DXC:GA 0% and CG:DXC:GA 0.25%), but the addition of GA up to 0.75% determined a shift to higher wavenumbers, indicating an increase in the association by hydrogen bonding (1544 cm^−1^ for CG:DXC:GA 0.5% and CG:DXC:GA 0.75%). The addition of 1% GA in the collagen sponges with DXC determined a major decreasing at 1532 cm^−1^, which indicates that a high concentration of GA does not lead to a good cross-linking. For all the studied sponges, C–C and C–N bonds are evidenced by the amide III band recorded at ~1337 cm^−1^. Also, a C–N–C bond appeared at ~1237 cm^−1^, and the absorption band specific to the stretching vibration of C–OH (ν_C–OH_) was recorded at the same wavenumber (1081 cm^−1^) in the all types of sponges with active substance and GA, even if the concentration of GA varied from 0% to 1%.

According to Table 3, the intensity ratio of the bands A_OH_/A_I_ (A_~3300_/A_~1630 cm^−1^_) indicates that the hydrolysis degree increased by the addition of OTC, reaching a maximum for 0.25% GA, while DXC presence did not increase the hydrolysis degree.

High values for the intensity ratio of the bands A_I_/A_OH_ indicate a high degree of cross-linking [46].

The results obtained for Δν (Equation (4)) suggest that the denaturation process of the α-helix chain in the collagen molecule is not present during sample preparation, except for the collagen sponge with DXC cross-linked with 1% GA, where Δν > 100. This can be explained by the high amount of GA that does not react with the collagen. 

Comparing Δν value and the ratio of A_I_/A_OH_ corresponding to the collagen sponge with no added GA or drug (collagen sponge CG:GA 0%) with the values of the collagen sponge with GA and with OTC or DXC, the following can be observed.

The addition of OTC into the collagen sponge composition determined a slight decrease of the cross-linking degree, but it remains over two, and as the content of GA increased, the cross-linking degree increased and, in addition, Δν decreased.

The addition of DXC led to an increased cross-linking degree, and the addition of GA also determined an increase of the cross-linking degree, but Δν increased more, over 100, for the sponge CG:DXC:GA 1%, suggesting that a denaturation process occurred.

Since the sponges obtained should have a good cross-linking degree, do not cause collagen degradation, and do not contain a high percentage of cross-linking agent, it can be concluded that the best results are registered for CG:OTC:GA 0.5%, CG:OTC:GA 0.75%, CG:DXC:GA 0.5%, and CG:DXC:GA 0.75%.

The UV/Vis/NIR spectra for collagen sponges with antimicrobial drugs, and with or without the cross-linking agent (Figure 4) and the data collected in Table 4, indicate the specific absorption bands that demonstrates the contribution of each component.

According to Figure 5, the presence of the drug determined an increase in water absorption. Also, the swelling behavior of the sponges depends on the type of antimicrobial drug. Thus, DXC led to higher values of swelling degree comparing with those obtained when OTC was used. The addition of small quantities of GA (0.25%) in collagen sponges with OTC determined an increase in water absorption, while in the case of DXC, the water absorption decreased. When GA concentration increased from 0.5% to 1% in collagen sponges with OTC, the high association degree by hydrogen bonds reduced the water absorbance, and these results were in line with those obtained from spectral analysis. A similar behavior was observed for collagen sponges with DXC with one exception for 1% GA. It should be noted that cross-linked collagen sponges with OTC added absorb less water over a longer period of time (5 h), while cross-linked collagen sponges with DXC added absorb more water in a shorter period of time (4 h).

From the enzymatic degradation point of view (Figure 6 and Figure 7), the non-cross-linked collagen sponge with no antimicrobial drug (sponge CG:GA 0%) was totally digested after 1 h of incubation with collagenase. The non-cross-linked collagen biomaterials treated with the antimicrobial drug were completely degraded after 24 h for the sponge with DXC, and after 72 h for the sponge with OTC. The collagen sponges with the antimicrobial drug and GA were totally degraded after different intervals of time depending on the cross-linking degree. It can be noticed that for higher GA concentrations (1%), the sponge with DXC was completely digested in the first 8 h, and the sponge with OTC was completely digested in 72 h. In addition, the collagen sponges containing 0.5% or 0.75% GA and treated with DXC were degraded between 70–74% after 96 h, while the collagen sponges with 0.5% or 0.75% GA treated with OTC were degraded between 55–63% after 96 h. Total degradation for the collagen sponges with 0.5% or 0.75% GA treated with DXC or OTC was obtained after 168 h. It is known that preservation of material structural integrity is one of the main properties of the collagen biomaterials to ensure an easy removal from the treated area. Thus, the collagen degradation values revealed that for both antimicrobial drugs, the cross-linking with 0.5% GA leads to formulations that best act as inhibitors of enzymatic activity, improving the biostability of the designed sponges, and the enzymatic degradation results indicating a better inhibition of collagenase in the case of oxytetracycline for this level of glutaraldehyde.

The antimicrobial activity investigation of the tested sponges was performed on *Escherichia coli*, *Staphylococcus aureus*, and *Enterococcus faecalis* (Table 5 and Table 6). We selected these test bacteria because in dentistry, *Enterococcus faecalis* has been found to be associated with chronic periodontitis [48] and failed root canal treatments involving chronic apical periodontitis [49]. At the same time, the transmission of *Staphylococcus aureus* in the dental setting is becoming more common [50,51]; *Escherichia coli* bacterium is not commonly found in root canals, but some studies found this microorganism in root canals with periapical lesions [52,53].

The results presented in Figure 8 demonstrate that in most cases for a concentration of 1% GA into the CG–OTC sponges, the inhibition percent of bacteria growth decreases. On the tested samples, the best antibacterial effect was demonstrated against *Escherichia coli* (Gram-negative bacterium), while the lowest was demonstrated for the bacterium *Enterococcus faecalis* (Gram-positive bacterium). As it is well known, Gram (+) bacterium was much more difficult to destroy due to the more complex structure of the cell wall (several layers of peptidoglycan), which is than Gram (−) bacteria whose cell wall is relatively thin and composed of a single peptidoglycan layer [54]. OTC, being a broad-spectrum antibiotic that is active against a wide variety of bacteria, works by interfering with the ability of microorganisms to produce essential proteins. Without these proteins, the microorganism cannot grow and multiply. Therefore, OTC stops the spread of the bacterial infection and the remaining bacteria are killed [5]. The results indicate that 0.5% and 0.75% GA represent the optimal content of the cross-linking agent for the CG-OTC sponges. In the case of CG-DXC sponges, the optimal content of GA is 1% (Figure 9). These data revealed that the new collagen sponges with OTC present a better antibacterial activity against Gram-positive and Gram-negative bacteria than systems with DXC. The action of DXC, which is also a broad-spectrum antibiotic, consists of inhibiting the synthesis of bacterial proteins by binding to the 30-S ribosomal subunit, and stops bacterial growth, giving the immune system time to kill and remove the bacteria [55].

From the antibiotics release profiles (Figure 10 and Figure 11), both antibiotics showed similar kinetic models. In all the cases, the drug delivery process could be divided in two different stages: an initial period of rapid release corresponding to the first 60–90 min, followed by a relatively slow drug release during the next 9 h of experiments for CG–OTC sponges, and 7 h for CG–DXC sponges. These delivery patterns might be due to the immediate desorption of the free drug non-entrapped in the spongious matrices, followed by a reduced release through diffusion of the antibiotic retained in the polymer matrix after gradual release medium penetration in the sponge structure [31]. The aforesaid drug-release characteristics are desired for both prophylactic and treatment of an infection associated with different dental procedures when it is necessary to achieve a high antibiotic concentration in a short period of time, followed by a sustained release to avoid further bacterial invasion or proliferation [56]. 

For both antibiotics, the burst release effect is more pronounced for smaller concentrations of GA. In the case of sponges with OTC, the drug-released percentage in the first 90 min is 50.07% for CG:OTC:GA 0% and 47.65% for CG:OTC:GA 0.25%, while for the sponges with DXC, the drug released percentage in the first 60 min is 59.62% for CG:DXC:GA 0% and 55.68% for CG:DXC:GA 0.25%. The increase of GA amount between 0.50–1.00% minimized the fast release of about 11.5–24.0% for CG–OTC sponges and about 24.50–38.90% for CG–DXC sponges compared to the corresponding non-cross-linked sample. It is obvious that the burst release effect is more evident for DXC-loaded collagen sponges. 

After 10 h of experiments, the cumulative released percentage of OTC ranges from 56.59% to 75.83%, while for the samples with DXC, the percentage varies between 61.75–78.80% over a period of 8 h. The sponges cross-linked with 0–0.5% GA led to a higher percentage of antibiotics released in contrast with samples with a high cross-linking degree (Table 7 and Table 8), with the increase being about 1.08 to 1.34 times for CG–OTC matrices, and about 1.09 to 1.27 times for CG–DXC matrices. It can be noticed that the cumulative released drug percentage in 8 h is higher for DXC-loaded collagen sponges than for OXC-loaded collagen sponges in 10 h.

Also, in the case of CG–OTC sponges, the percentages of released drug were lower compared to the CG–DXC sponges cross-linked with the same amount of GA, and these results were in line with the ones obtained for the water absorption (Figure 5). The results indicate that the cross-linking degree and the type of antibiotic are the major parameters that influence the drug release kinetics from the designed sponges.

The correlation coefficient values obtained for the power law kinetic model (Equation (3)) are presented in Table 7 and Table 8. These values are significantly higher than the ones obtained for its particular cases, namely the Higuchi and zero-order models. The release exponent values ranged between 0.267–0.288 for CG–OTC sponges, and from 0.181 to 0.318 for CG–DXC sponges respectively, suggest a typical non-Fickian drug-release mechanism. These results are in line with our previous works [24,30,31,32,33], which also used collagen sponges cross-linked with glutaraldehyde as carriers for different drugs, as well as with other author works [57,58].

Even if the two drugs belong to the same class of tetracycline and consequently their chemical structures are quite similar, the different behavior is attributed to the different bonds established between collagen, GA, and drugs.

Figure 12 shows that untreated or treated collagen sponges (CG:GA 0% and CG:GA 0.5% sponges) were non-cytotoxic and biocompatible, and the cell viability recording values were 118–116%, higher than that of the control culture at 24 h when cell proliferation was stimulated. After 48 h, the cell viability values of the sponges decreased to 96–93%, but with an obvious non-cytotoxic and biocompatible character. In the case of the CG:OTC:GA 0.5% sponge, compared to the control culture, the cell viability was 83% at 24 h and 96% at 48 h. This confirms the non-cytotoxic and biocompatible nature of the CG:OTC:GA 0.5% sponge.

Hematoxylin–eosin staining carried out on cell culture is designed to help in the cell morphology interpretation obtained after the cytotoxicity experiment. Experimental observations of the cellular morphology resulting from this examination are as follows.

In the control culture well, a morphological aspect specific to the NCTC cell line with round and polygonal fibroblast cells was observed. After 48 h from seeding, the cell culture was close to the confluence (90% confluence) (Figure 13a).

When the cell culture was treated with hydrogen peroxide 0.003% (M+), a strong cytotoxic effect occurred, which led to the total destruction of the cell substrate; the attached cells are round, the cell membrane is damaged, and both the cytoplasm and intracytoplasmic granules are missing from the cell constitution (Figure 13b).

Cell cultures in contact with a non-cross-linked collagen sponge (CG:GA 0%) and with a cross-linked collagen sponge (CG:GA 0.5%) showed a specific morphology to the NCTC line; the culture was subconfluent (~80%) and the cell density was close to that of the control culture, but the cell metabolism was lower than that of the control culture. The images (Figure 13c,d) reflect the non-cytotoxic character of the sponges. 

The cell culture in contact with the collagen sponges treated with GA and with OTC (CG:OTC:GA 0.5%) showed a specific morphology to the NCTC line, with round and polygonal fibroblast cells with homogeneous aspects. The cell density was close to that of the control culture, with the cell culture evolution being more advanced compared to the culture in contact with CG:GA 0% and CG:GA 0.5% sponges. The biocompatibility of the CG:OTC:GA 0.5% sponge was evident (Figure 13e).

## 5. Conclusions

In this paper, we presented a comparative study concerning the properties of new sponges prepared and characterized by numerous methods to provide their use as carrier supports in biomedical applications. The ability to absorb water depending on the antimicrobial drug or chemical cross-linking agent concentration is in line with the data obtained from spectral analysis. The drug release kinetics could be controlled by the cross-linking degree, with the GA concentration increase determining the decrease of the released drug percentage, and by the nature of the antibiotics, with an extended release being recorded for oxytetracycline-loaded collagen sponges. The collagen degradation values revealed that both 0.5% GA and OTC presence into the CG:OTC:GA 0.5% sponge composition ensure the best inhibitors of enzymatic activity, improving the biostability of the sponge. For this formulation, the non-cytotoxic and biocompatible nature was confirmed. Considering that a quite fast antibiotic release just after a dental procedure followed by a slower and gradual delivery during a longer period are targeted for local anti-infective dentistry therapy, the collagen sponge with 0.5% GA could be a viable topical support for oxytetracycline, with good results in the prevention/treatment of the infection at the application site, favoring tissue regeneration.

## Figures and Tables

**Figure 1 pharmaceutics-11-00363-f001:**
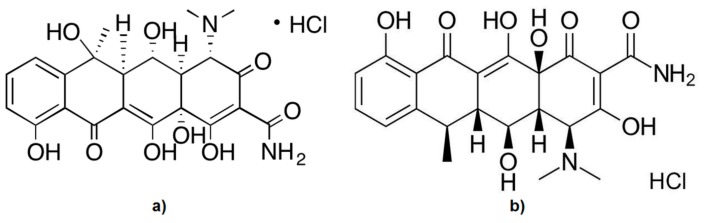
Chemical structure: (**a**) oxytetracycline hydrochloride; (**b**) doxycycline hydrochloride.

**Figure 2 pharmaceutics-11-00363-f002:**
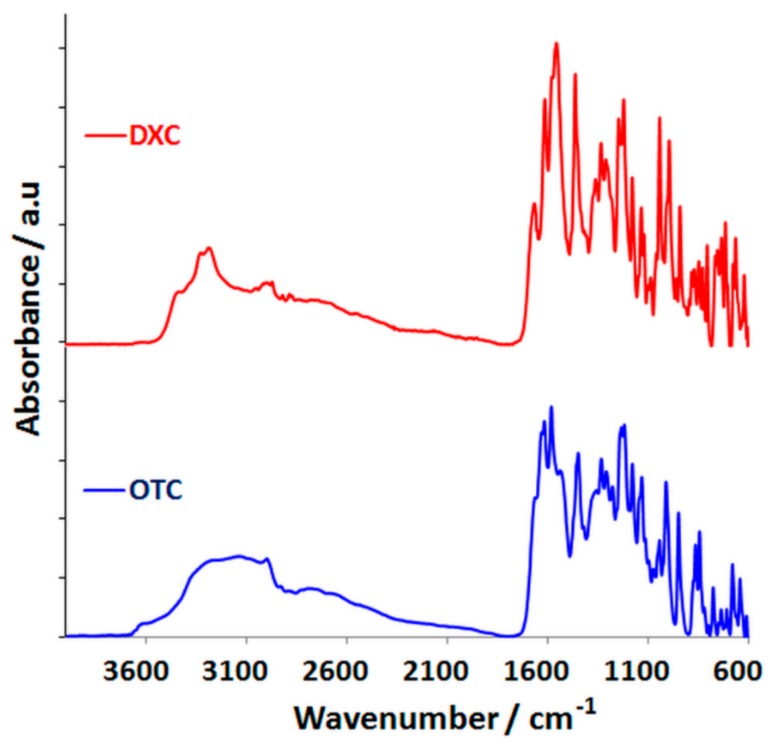
The IR spectra of OTC and DXC powder.

**Figure 3 pharmaceutics-11-00363-f003:**
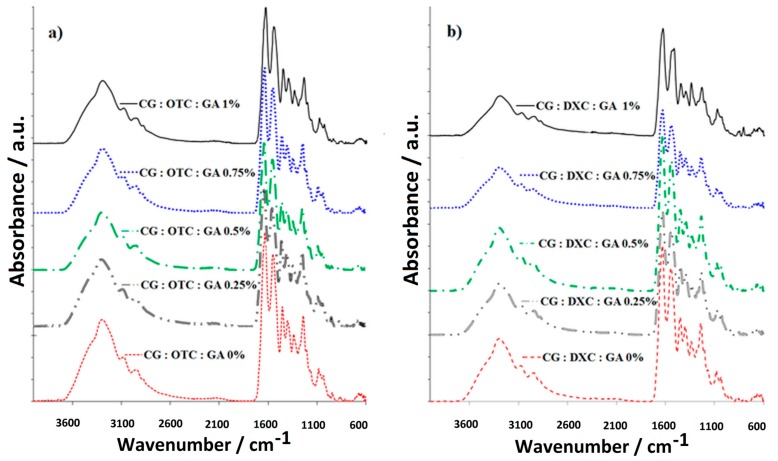
The IR spectra of collagen sponges with different GA concentrations and with antimicrobial drug (**a**) OTC or (**b**) DXC.

**Figure 4 pharmaceutics-11-00363-f004:**
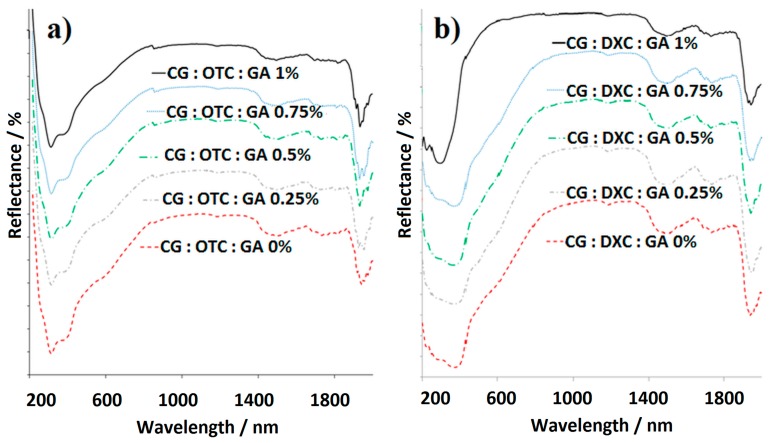
The UV/Vis/NIR spectra of collagen sponges with different GA concentrations with (**a**) OTC or (**b**) DXC.

**Figure 5 pharmaceutics-11-00363-f005:**
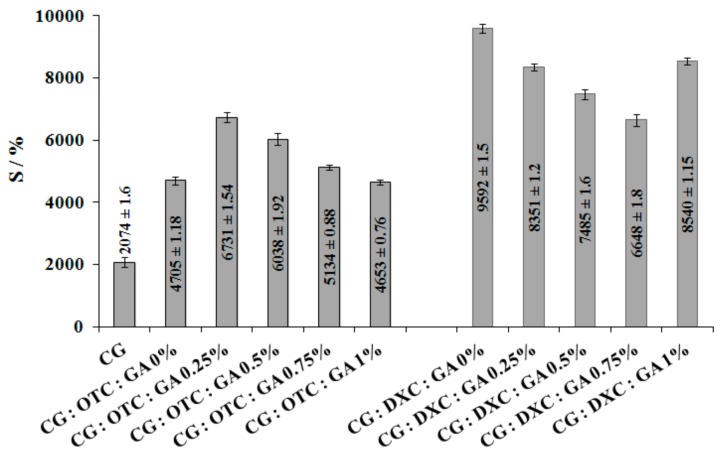
The swelling capacity of the CG–OTC and CG–DXC sponges with different GA concentrations.

**Figure 6 pharmaceutics-11-00363-f006:**
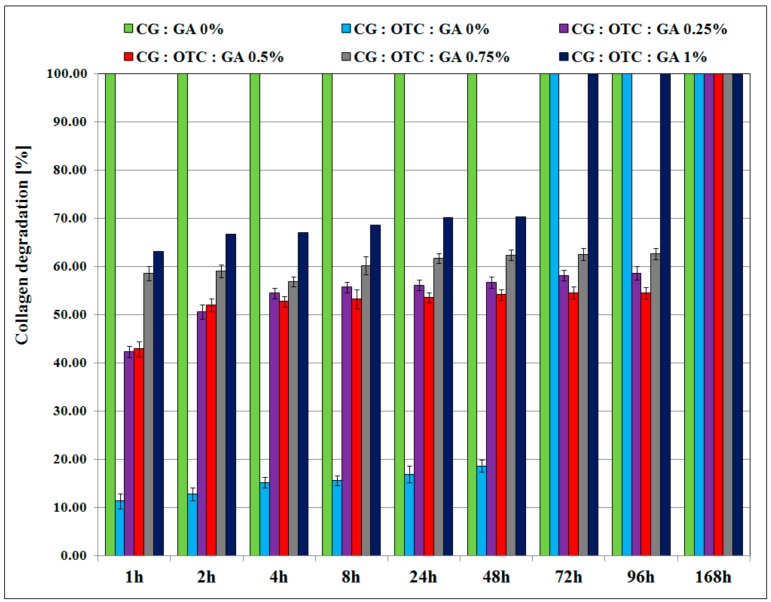
The percentage of collagen degradation from collagen sponges with or without OTC and different GA concentrations.

**Figure 7 pharmaceutics-11-00363-f007:**
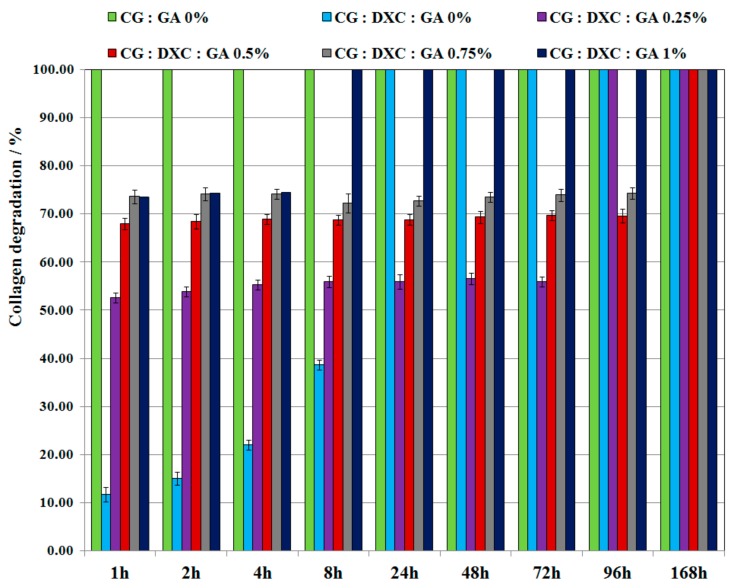
The percentage of collagen degradation from collagen sponges with or without DXC and different GA concentrations.

**Figure 8 pharmaceutics-11-00363-f008:**
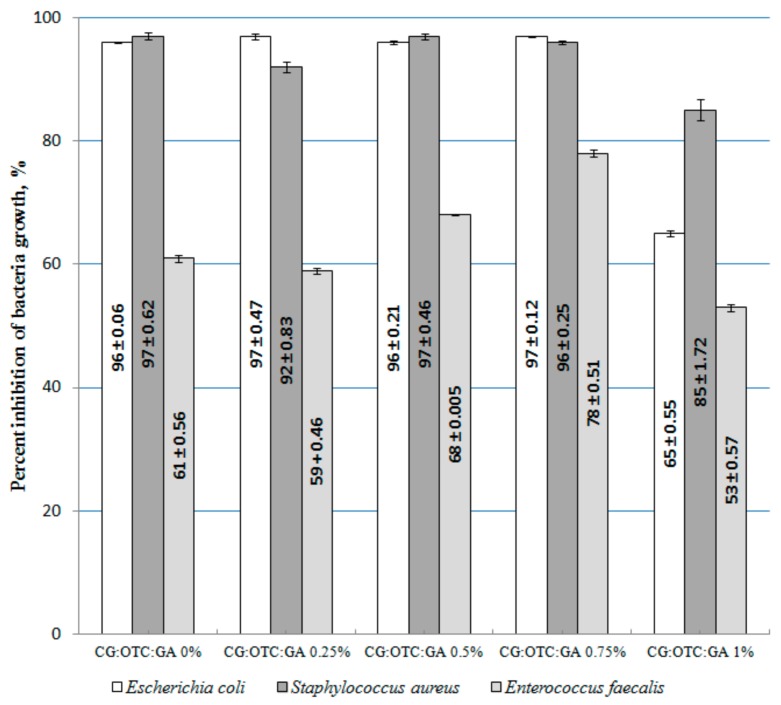
Inhibition percent of bacteria growth for the system with oxytetracycline.

**Figure 9 pharmaceutics-11-00363-f009:**
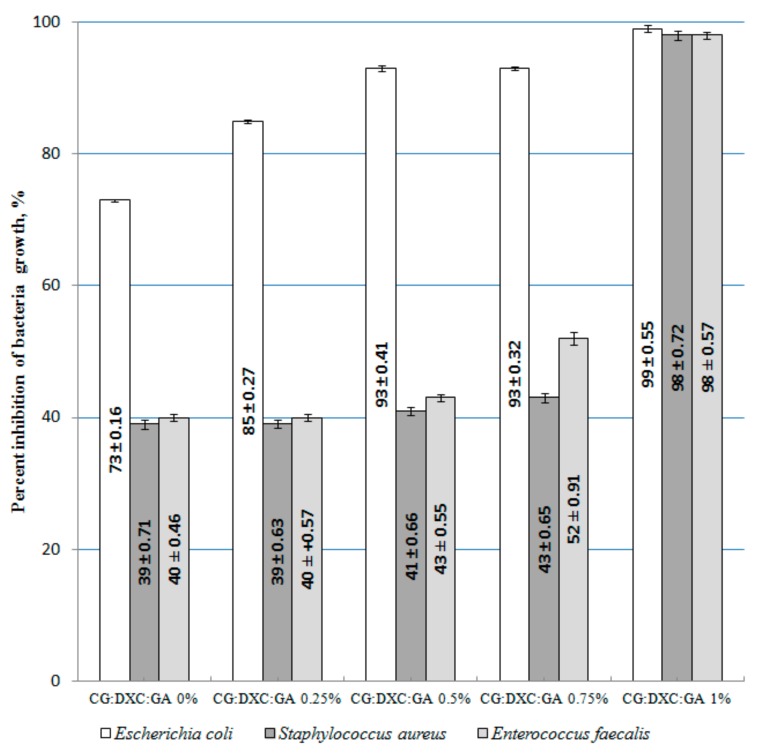
Inhibition percent of bacteria growth for the system with doxycycline.

**Figure 10 pharmaceutics-11-00363-f010:**
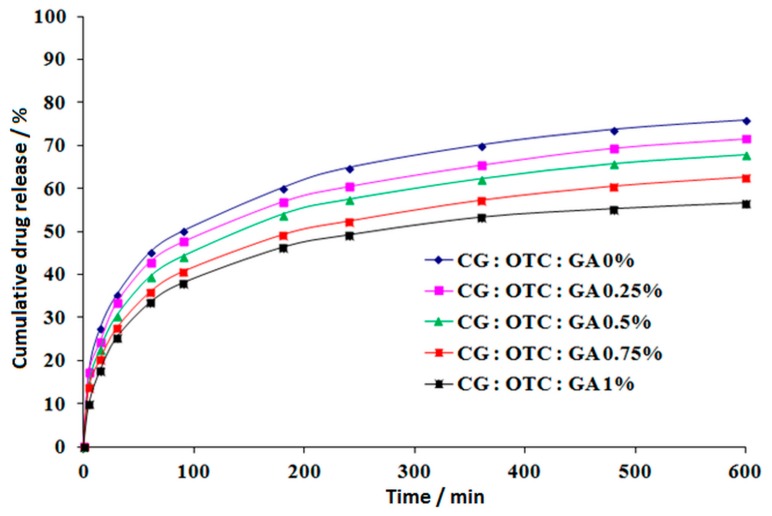
In vitro release of oxytetracycline from collagen sponges.

**Figure 11 pharmaceutics-11-00363-f011:**
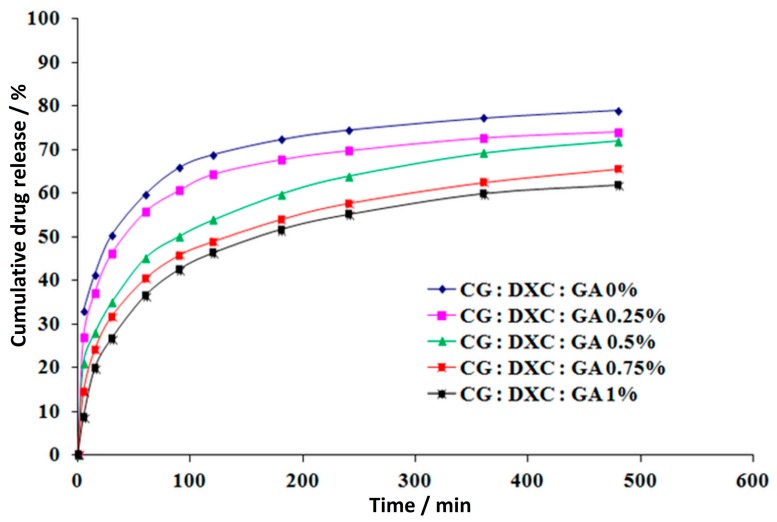
In vitro release of doxycycline from collagen sponges.

**Figure 12 pharmaceutics-11-00363-f012:**
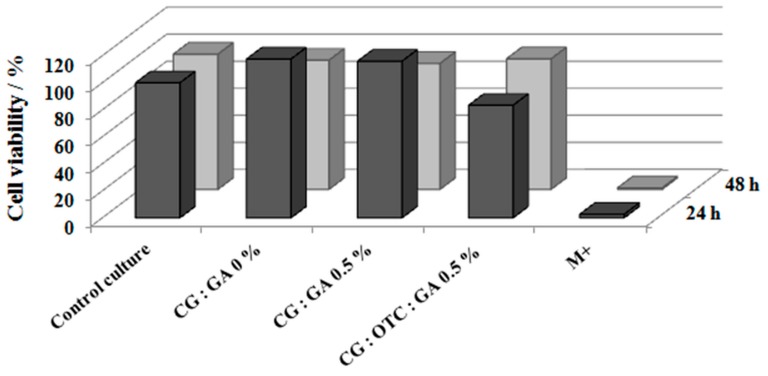
Cell viability of collagen sponges.

**Figure 13 pharmaceutics-11-00363-f013:**
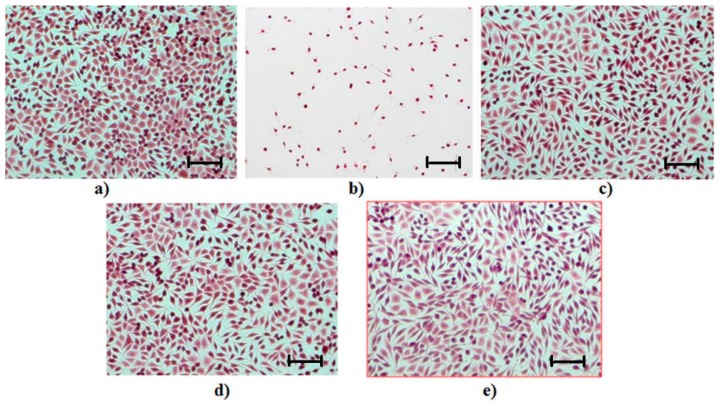
Image of cell morphology for: (**a**) the control culture; (**b**) the cell culture treated with 0.003% hydrogen peroxide, with the role of M+ (positive control); (**c**) the culture in contact with the CG:GA 0% sponge; (**d**) the culture in contact with the CG:GA 0.5% sponge; and (**e**) the culture in contact with the CG:OTC:GA 0.5% sponge. Scale bar: 50 µm.

**Table 1 pharmaceutics-11-00363-t001:** Composition of the CG–OTC and CG–DXC sponges with different GA concentrations. CG: collagen, DXC: doxycycline hydrochloride, GA: glutaraldehyde, OTC: oxytetracycline hydrochloride.

**Collagen Sponge**	**Gravimetric Ratio** **CG:OTC [w:w]**	**GA [%]**
CG:OTC:GA 0%	1:0.1	0
CG:OTC:GA 0.25%		0.25
CG:OTC:GA 0.5%		0.50
CG:OTC:GA 0.75%		0.75
CG:OTC:GA 1%		1.00
**Collagen Sponge**	**Gravimetric Ratio** **CG:DXC [w:w]**	**GA [%]**
CG:DXC:GA 0%	1:0.1	0
CG:DXC:GA 0.25%		0.25
CG:DXC:GA 0.5%		0.50
CG:DXC:GA 0.75%		0.75
CG:DXC:GA 1%		1.00

**Table 2 pharmaceutics-11-00363-t002:** IR frequencies of the functional groups in OTC and DXC.

OTC		DXC	
ν [cm^−1^]	Functional Group	ν [cm^−1^]	Functional Group
3145	ν_O–H_	3281	ν_O–H_
2997	ν_CH2_	2992	ν_CH2_
2925	ν_CH2_	2968	ν_CH2_
1616	ν_C=O_		
1579	δ_N–H_	1571	δ_N–H_
1538	ν_C=C aromatic_	1553	ν_C=C aromatic_
1331	δ_CH3_	1331	δ_CH3_
1070	ν_C–OH_	1084	ν_C–OH_

**Table 3 pharmaceutics-11-00363-t003:** Fourier transform infrared (FT-IR) spectra data of collagen sponges.

Collagen Sponge	A_OH_/A_I_	A_I_/A_OH_	Δν
CG:GA 0%	0.4111	2.4326	84
CG:OTC:GA 0%	0.4834	2.0689	82
CG:OTC:GA 0.25%	0.4918	2.0332	82
CG:OTC:GA 0.5%	0.4551	2.1973	86
CG:OTC:GA 0.75%	0.4484	2.2301	81
CG:OTC:GA 1%	0.4624	2.1627	81
CG DXC:GA 0%	0.4023	2.4859	91
CG:DXC:GA 0.25%	0.4158	2.4051	92
CG:DXC:GA 0.5%	0.4027	2.4835	87
CG:DXC:GA 0.75%	0.4166	2.4004	87
CG:DXC:GA 1%	0.3721	2.6876	102

**Table 4 pharmaceutics-11-00363-t004:** UV/Vis/NIR spectra data for collagen sponges with antimicrobial drug and different GA concentrations.

Collagen Sponge	λ_max_, nm	Band	λ_max_, nm	Band	λ_max_, nm	Band	λ_max_, nm	Band
CG:OTC:GA 0%	313	–CO–NH–	1185	ν_CH2_	1492	ν_OH as_	1942	δ_O–H_
CG:OTC:GA 0.25%	314		1190		1495		1954	
CG:OTC:GA 0.5%	314		1188		1496		1944	
CG:OTC:GA 0.75%	314		1188		1501		1953	
CG:OTC:GA 1%	314		1189		1497		1947	
CG:DXC:GA 0%	223		1183		1504		1937	
CG:DXC:GA 0.25%	223		1183		1503		1941	
CG:DXC:GA 0.5%	221		1181		1503		1945	
CG:DXC:GA 0.75%	224		1183		1492		1938	
CG:DXC:GA 1%	223		1181		1504		1948	

**Table 5 pharmaceutics-11-00363-t005:** Antimicrobial susceptibility of the microorganisms to oxytetracycline.

Microorganism	Concentration of Oxytetracycline (µg/mL)											
	400	200	100	50	25	12.5	6.25	3.125	1.56	0.78	0.39	0.195
*Escherichia coli*	S	S	S	S	S	S	R	R	R	R	R	R
*Staphylococcus aureus*	S	S	S	R	R	R	R	R	R	R	R	R
*Enterococcus faecalis*	S	S	R	R	R	R	R	R	R	R	R	R

Key: R—Resistant; S—Susceptible/Sensitivity.

**Table 6 pharmaceutics-11-00363-t006:** Antimicrobial susceptibility of the microorganisms to doxycycline.

Microorganism	Concentration of Doxycycline, (µg/mL)											
	400	200	100	50	25	12.5	6.25	3.125	1.56	0.78	0.39	0.195
*Escherichia coli*	S	S	S	R	R	R	R	R	R	R	R	R
*Staphylococcus aureus*	S	S	R	R	R	R	R	R	R	R	R	R
*Enterococcus faecalis*	S	S	R	R	R	R	R	R	R	R	R	R

Key: R—Resistant; S—Susceptible/Sensitivity.

**Table 7 pharmaceutics-11-00363-t007:** Correlation coefficients for Higuchi, zero-order, and power law kinetic models; kinetic parameters specific to the power law model and cumulative oxytetracycline released percentage.

Collagen Sponges	R (Higuchi Model)	R (Zero-Order Model)	R (Power Law Model)	Release Exponent	Kinetic Constant (1/min^n^)	OTC Released Percentage (%)
CG:OTC:GA 0%	0.9531	0.8445	0.9948	0.267	0.143	75.83
CG:OTC:GA 0.25%	0.9524	0.8438	0.9943	0.267	0.135	71.59
CG:OTC:GA 0.5%	0.9564	0.8496	0.9938	0.279	0.119	67.81
CG:OTC:GA 0.75%	0.9594	0.8551	0.9945	0.283	0.107	62.57
CG:OTC:GA 1%	0.9512	0.8394	0.9879	0.288	0.095	56.59

**Table 8 pharmaceutics-11-00363-t008:** Correlation coefficients for Higuchi, zero-order, and power law kinetic models; kinetic parameters specific to the power law model and cumulative doxycycline released percentage.

Collagen Sponges	R (Higuchi Model)	R (Zero-Order Model)	R (Power Law Model)	Release Exponent	Kinetic Constant (1/min^n^)	DXC Released Percentage (%)
CG:DXC:GA 0%	0.8784	0.7193	0.9919	0.181	0.273	78.80
CG:DXC:GA 0.25%	0.8891	0.7316	0.9891	0.196	0.235	74.07
CG:DXC:GA 0.5%	0.9514	0.8300	0.9966	0.262	0.148	71.96
CG:DXC:GA 0.75%	0.9525	0.8300	0.9932	0.280	0.122	65.48
CG:DXC:GA 1%	0.9575	0.8383	0.9865	0.318	0.093	61.75

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
