# Peer review of "Oxytetracycline versus Doxycycline Collagen Sponges Designed as Potential Carrier Supports in Biomedical Applications"

_pharmaceutics, 2019, doi:10.3390/pharmaceutics11080363_

Reviewer 1 Report

The work entitled “Oxytetracycline versus doxycycline collagen sponges as carrier supports in dentistry” offers a comparative study about the properties of new collagen sponges prepared and characterized by different methods. The work is well conducted and the data well discussed. Overall, the work is scientifically sound.

The English of the entire manuscript should be improved. There are important verb conjugation and grammar mistakes.

In detail:

Abstract: does not show the most relevant information acquired from the work, there are no major findings nor conclusions. This is most important. The abstract should provide a complete overview of the work conducted and here it is basically introduction and goals.

Introduction: it is clear and offers a complete overview of the subject. The goal is also well presented.

Materials and Methods: well presented and most of the times very complete. However, the antimicrobial section is a bit difficult to follow. You should provide more detail and not just refer to other studies.

Results: it would be important to add statistical analysis in the histograms, particularly in Figure 8 and 9.

Discussion: very complete, well-structured and scientifically sound. It would be important to add more interaction between the results and not just present them individually but aside from that it was easily understood and scientifically sound.

Reviewer 2 Report

The manuscript is to compare the collagenase digestion rate, drug kinetic release profile and in vitro biocompatibility of oxytetracycline and doxycycline containing collagen sponge cross-linking with glutaradehyde. The authors provide lots of interesting viewpoints. However, I have some suggestions before the manuscript to be publish.  

Since the main purpose of this work is to comparative study of the drug kinetic release profile of two antibiotics, therefore, the drug kinetic release analysis is the most important section in the manuscript. Every mathematical model has its unique characters, such as assume conditions and boundary conditions. Thus, firstly the authors should consider the designed experiment environments meet which one math model requirements, and then analyze the curve fitting results. Using experiment data to find out best fitting math model is not right approaching method. And the authors should compare the drug release profile and models with others similar works, which are collagen sponge cross-linking with GA used as drug carriers, in the discussion section. 

The drug release experiment is determined in non-collagenase or collagenase containing solution, please indicate it. How about the other condition (non-collagenase or collagenase) on drug release because obviously degradation at 1 h ? 

Basically, materials pore size and porosity are two key important parameters of drug carriers. The authors should compared the two factors with other similar works to highlight the specific of their fabrication. 

The title “Oxytetracycline versus doxycycline collagen sponges as carrier supports in dentistry”, however, I do not see any experiments is related with dentistry in the whole manuscript. Bacterial infection is a general phenomenon. 

In introduction, the authors should indicate what the study is important and what they can provide useful messages that did not obtain from previous works rather than general statements.

The entire text need to read one more time carefully. Some section has to simplify or delete it especially in materials and method part. 

Line 71_in my knowledge, the break down collagen mechanism is different between mammalian collagenase and bacterial collagenase. Therefore, this section is not very correct, and I cannot find any relationship with this topic of manuscript.  

Line 96_“Freezing” at 55 °C for 10 min …..”, I cannot understand how to freeze at 55 °C.

Line 119_why the collagen materials have to immersed in PBS for 24 h before added collagenase? How to determine the duration time? And how to “squeezed” the samples?(line 121)?

Line 160_there are numerous models used to analyse release kinetic drug from materials. The authors have to explain why choose power law kinetic model to explain their data in this work. 

Line 173_”dermal fibroblasts of mice cell line NCTC (clone 929)”, please indicate the cell line providing source and Lot (or ATCC) number.

Line 192_For cell morphology examination_ this section is fameless.

Line 209_”intensity ratio of the bands AOH / AI (A~3300 / A~1630 cm-1) it can be appreciated that hydrolysis degree”, please explain the statement and add suitable citations. 

Line 331_ “High values of the intensity ratio of the bands AI / AOH indicate a high degree of cross-linking”, the authors have to explain what is meaning of this sentence in the text. For example, based on table 3, DXC could cross-linking CG, and GA cannot cross-linking CG+DXC until concentration up to 1%.

Line 333_ “The results obtained for Δv (Equation (4)) suggest that the denaturation process is not present”, what is meaning of this sentence? Is the collagen denaturation is not present within the sample? If it is true, how the DXC and 1% GA will induce collagen denaturation? And please add suitable citations. 

Line 248_ based on the statements in materials and method section, “The collagen sponges ….immersion in distilled water for different periods of time up to 5 h at 25 °C. …collagen sponges were removed and weighed without dripping (Wf)”, so what is the time point of figure 5? And please Statistical analyses these data.

Line 451_ “comparative study of the properties of new sponges prepared… by different methods to …”, what is different “method”? 

Reviewer 3 Report

Authors have fabricated OTC and DXC loaded collagen sponge with and without GA cross-linking as a polymeric support for prevention / treatment of the infection in dental applications. The interactions between OTC/DXC with collagen, the resistance to collagenase digestion, and antimicrobial effect have been systematically tested for different prepared sponges. This work is deserved to be published in Pharmaceutics after minor revisions.

- It is not safe to say the material is favoring the tissue regeneration. There is no in vivo experiment (just in vitro cell experiments).

- In Fig. 1, 2: Please use solid lines with different colors (dash or dot lines sometimes are hard to find the peaks)

- In Table 1, GC : OTC,  GC : DXC. Should them to be CG (collagen), instead of GC?

- Fig. 5-9, please keep the bar plot as simple and clarity as possible. It looks complicated in bar plots by the combination of line-type, fill-type and mask patterns. 

Author Response

Round  2

Reviewer 1 Report

The authors have addressed all the reviewers comments and, as such, the work now meets the standards of this journal. 

I recommend its publication.

Author Response

Thank you for your valuable comments.

Reviewer 2 Report

The authors have given very detail responses and amended their manuscript according my suggestions. However, I still not be convinced on some points.   

-The authors mentioned that the present work was an in vitro studied the physical and chemical properties of prepared sponges, and suggest it may have potential using in dental infection treatment. However, there are no any experiments were related with this issue in the whole manuscript, although the drug used in the study are general used in treated dental infection. I still think the title should amend based on present data. 

-Introduction: collagen has been widely used as drug carriers with several methods such as lyophilized, leached out, and electrospun etc., and GA is a common used cross-linking agent. In introduction section, the authors should point out what is important or different points between their work with other previous literatures which also used GA as the crosslinking agent to prepare drug-containing collagen carriers, rather than general introduction of materials. 

- Line 89-95 only describe results came from references, for my opinion, the authors should compare the pore size and porosity with other similar works to highlight the specific of their fabrication process or compare the influences of OTC and DXC on the pore size and porosity since the two drug caused different release profile.  

-Results: It would be important to add more explanations about the results not just only present them individually. Such as line 274-276“UV/Vis/NIR spectroscopy was used to depict the major components of the collagen-based biomaterials …. the UV/Vis/NIR spectra are presented in Figure 4”, there are no any describing about Fig 4. Readers are hard to understand what is the meaning of fig.4 because all of curve look very similarly. Authors have to lead readers to understand the results, so authors can consider moving some paragraphs from discussion section to results section.  

-Discussion: Title of this manuscript is “oxytetracycline versus doxycycline collagen sponges”, and the authors did lots of experiments to show the different physical properties of the two drug containing collagen carriers. Therefore, in discussion section, I think authors have giving a reason why the two drug caused different results rather than only describing results. 

- “According to the IR data (Figure 3 and Table 3), the absorption band attributed to νNHasand νOHasat 3294 cm-1for …”, I cannot find any mention about “νNHasand νOHasat 3294 cm-1” on Figure 3 and Table 3. Please indicate every specific wavenumber, which are showed in discussion section, on figure. 

-Line 436-438: “the collagen degradation values ….(0.5% GA) and antimicrobial drugs may act as inhibitors of enzymatic …”. To my knowledge, all of cross-linked collage will slow down collagenase degradation rate. That is until numerous researchers would like to develop new method to improve collagen Xlinking degree. Therefore, the statement is not very properly .    

Author Response

Thank you for your comments. Please find our responses in the attached PDF.

Round  3

Reviewer 2 Report

The authors have amended their manuscript point-by-point based on my reviewer report. Before the manuscript is published, I suggest the authors have to include “Statistical analysis” in Materials section because the major aim of the present study is compared performance of two drug carriers. I am very sorry for missing the important point in my previous report.  

Author Response

Answers to R2 (Round 3)

Thank you very much for your suggestions and for helping us to improve the manuscript! The authors included “Statistical analysis” (by “Track changes”) in Materials and Methods section as you suggested.  

Please, see new Lines 284-288.

Statistical analysis

All experiments performed on the collagen sponges studied were carried out in triplicate, and the data obtained were analyzed for statistical significance using analysis of variance (one-way ANOVA) and Tukey test to determine the significant differences between means. Significant differences were considered for P values < 0.05. The results were reported as samples
mean ± standard deviation (SD) of independent replicates.
